# The Gut Microbiota in Celiac Disease and Probiotics

**DOI:** 10.3390/nu11102375

**Published:** 2019-10-05

**Authors:** Richa Chibbar, Levinus A. Dieleman

**Affiliations:** Division of Gastroenterology, Department of Medicine, University of Alberta, Edmonton, AB T6G 2X8, Canada; chibbar@ualberta.ca

**Keywords:** Celiac disease, probiotics, gut microbiota

## Abstract

Celiac disease (CeD) is an immune-mediated enteropathy, and unique in that the specific trigger is known: gluten. The current mainstay of therapy is a gluten-free diet (GFD). As novel therapies are being developed, complementary strategies are also being studied, such as modulation of the gut microbiome. The gut microbiota is involved in the initiation and perpetuation of intestinal inflammation in several chronic diseases. Intestinal dysbiosis has been reported in CeD patients, untreated or treated with GFD, compared to healthy subjects. Several studies have identified differential bacterial populations associated with CeD patients and healthy subjects. However, it is still not clear if intestinal dysbiosis is the cause or effect of CeD. Probiotics have also been considered as a strategy to modulate the gut microbiome to an anti-inflammatory state. However, there is a paucity of data to support their use in treating CeD. Further studies are needed with therapeutic microbial formulations combined with human trials on the use of probiotics to treat CeD by restoring the gut microbiome to an anti-inflammatory state.

## 1. Introduction

Celiac disease (CeD) is an immune mediated enteropathy triggered by ingestion of gluten in genetically predisposed individuals carrying human leucocyte antigen (HLA) DQ2 or DQ8. The current mainstay of treatment is adherence to a strict gluten-free diet (GFD) [1,2,3,4]. The global prevalence of CeD has been increasing worldwide, and in the North America the prevalence increased five-fold mirroring the increase in Inflammatory Bowel Disease (IBD) [5,6,7,8,9]. 

The key genetic elements (HLA-DQ2 and HLA-DQ8), the auto-antigen (tissue transglutaminase 2), and the external trigger (gluten) causing CeD are fairly well established. HLA-DQ2/8 is a common genotype and is noted in approximately 35% of the population, and approximately 3% of individuals develop CeD upon gluten exposure, suggesting a role for other factors in the development of CeD [10,11,12,13]. Growing evidence suggests that gut microbiota is closely related to digestive tract diseases, including CeD [14,15]. The gut microbiota plays a crucial role in mucosal differentiation, function, energy generation, and modulation of innate and adaptive system [16,17,18]. Alterations, probably due to improved hygiene and reduced exposure to various microorganisms, have been implicated in the pathogenesis of IBD [19]. Similarly, changes in the gut microbiome in HLA-DQ2/8 individuals can alter processing of gluten in the intestinal lumen, affect intestinal barrier, innate or adaptive immune responses, and may cause or contribute to gluten sensitive enteropathy [20]. As innovative therapies are developed, there is a paucity in understanding the role of the gut microbiota in CeD, specifically pathogenesis and clinical course. It is also unclear if modulation of the gut microbiome alters the natural history of CeD. In this review, we will discuss the association of gut microbiota in CeD. 

## 2. Pathogenesis of Celiac Disease:

In genetically susceptible patients, the pathogenesis of CeD starts with the ingestion of gluten-containing foods, which are incompletely digested in the intestinal lumen into potentially immunogenic gluten derived peptides (10- > 30 amino acids in size). Immunogenicity of these peptides varies, with 13-, 19- and 33-mers being more immunogenic and triggering immune response associated with CeD. These peptides contain six copies of different epitopes to which most individuals react [20,21,22]. Gliadin peptides containing nine or less amino acids have reduced immunogenicity [23]. Some of the commensal duodenal microbiota also have peptidase activity and break gliadins into smaller peptides [24,25]. 

To initiate the immune response, peptides translocate to the lamina propria by the paracellular route that involves the protein ‘zonulin’. Gliadin peptides bind to chemokine receptor, C-X-C motif chemokine receptor 3 (CXCR3) on epithelial cells, upregulate zonulin, and disassemble tight junctions leading to increased permeability [26,27,28]. Another pathway is transcellular, mediated by secretory immunoglobulin A (IgA) with the help of transferrin receptor (CD71) expressed on luminal surface of epithelial cells [29]. In the lamina propria, intestinal tissue transglutaminase (tTG) reacts with gliadin peptides to deaminate them to negatively charged glutamic acid residues that are highly immunogenic. These residues are recognized and processed by the HLADQ2 and HLA DQ8 bearing antigen presenting cells. The deaminated peptides and tTG complex activate CD^+^ T cells to generate antibodies against gliadin and tTG [30]. HLA-DQ2 and DQ8 variants enhance immune cell activation and autoimmunity by binding more tightly to gliadin peptides, thus accounting for 50% of genetic susceptibility [31]. Though non-HLA variants also regulate the structure and function of immune cells, it modestly increases the risk of CeD [32]. 

Innate immunity has an initial role in the development of CeD. Ingestion of gluten containing foods increase Interleukin-15 (IL-15) production causing polarization of dendritic cells, altering T-cell receptor-alpha beta intraepithelial lymphocytes (IELs) in the epithelium and damage to intestinal tissue [33,34]. Dysregulated interferon (IFN)-γ expression stimulates natural killer (NK) cells, CD^+^ T cell, and dendritic cell activation. The typical immune systems response is neutrophil infiltration and IL-8 release from the epithelium and immune cells [34]. Gliadin stimulates macrophage production of TNF-α, IL-8, RANTES, IL-1β, and nitric oxide. Alpha-amylase trypsin inhibitors also stimulate innate immunity through Toll-like receptors (TLR), myeloid differentiation factor-2 (MyD88), and CD14 complex [34]. Genome-wide association studies identified additional 39 non-HLA loci involved in immune function and confer CeD risk. Some of these non-HLA loci also regulate bacterial colonization and sensing [32]. Pathogenic bacteria associated immunogenicity is dependent on TLR transmembrane proteins. After recognition of pathogen, they activate innate immune system. Normal intestinal commensal bacteria do not activate immune system due to downregulation of TLR. Thus, there are similarities in activation of the. innate immune pathway in both, during invasion by pathogenic microorganism or autoimmunity by gliadin peptide due to loss of self-tolerance, as in both states there is an increased expression of TLR, release of pro-inflammatory cytokines and induction of NF-κβ. Increased TLR4 and TLR2 expression is also associated with both Inflammatory Bowel Disease (IBD) and CeD, implicating dysbiosis in disease pathogenesis [35,36,37,38]. Dysbiosis may affect autoimmunity by modulating the balance between commensal and pathogenic microorganisms and the host immune response, as discussed later.

In CeD the adaptive immune response is triggered by antigen-presenting cells (APC) that transport gluten peptides to CD4^+^ T cells, resulting in increased production and release of pro-inflammatory cytokines. In addition, increased production of metalloproteases and keratinocyte growth factor by stromal cells generate anti-gliadin and anti-tTG antibodies [39]. The response to gliadin is a Th1 driven process, while Th17 cytokines increase suggest that it also has a role in the development of CeD. Th17-mediated immune response is associated with alerted T-regulatory cell populations, which are also increased in active CeD. Th17 cells are regulated by the gut microbiota and also protect the host from infection, as well as other toxic molecules such as deaminated gliadin peptides [39].

## 3. Dysbiosis in Celiac Disease:

Approximately trillions of microorganisms inhabit our gut and contribute to normal bowel functions, including metabolic regulation and immune homeostasis [16,17,18,40]. The gut microbiota composition is established early in life and remains fairly constant throughout life in symbiotic tolerance with the host. Three bacterial phyla: *Firmicutes, Bacteroides* and *Actinobacteria* are the major components of the gut microbiota [41]. Dysbiosis is the imbalance of protective and pathogenic microbes in the host. It is typically caused by atypical microbial exposures, diet changes, antibiotic/medication use, and host genetics [40]. Initially, increased association of rod-shaped bacteria was reported in small bowel biopsies of active and inactive CeD patients [42]. Subsequently, in both stool cultures and duodenal biopsies reported an increased abundance of gram negative organisms, *Bacteroides, Clostridium*, *E.Coli* in CeD patients compared to healthy adults [43,44,45]. The concept of dysbiosis as risk factor for CeD was further strengthened by Swedish CeD epidemic study which also found higher numbers of rod-shaped bacteria (*Clostridium* spp., *Prevotella* spp., and *Actinomyces* spp.) in small bowel mucosa of CeD patients [46]. Since then there are several studies on fecal samples and duodenal mucosa using various techniques including 16SrRNA gene sequencing reporting similar results [47,48,49,50]. However, most of these studies are descriptive, some with patients on GFD or with gluten diet (GD) or symptomatic even on GFD. From these studies it is difficult to determine whether an altered gut microbiota is a cause or consequence of CeD, as GD and GFD can also modulate gut microbiota. Overall most of the duodenal biopsies from CeD patients compared to healthy subjects showed dysbiosis and revealed an increased number of Gram-negative bacteria, *Bacteroides, Firmicutes, E. Coli, Enterobacteriaceae, Staphylococcus,* and a decrease in *Bifidobacterium*, *Streptococcus*, *Provetella* and *Lactobacillus* spp. The studies of fecal samples and duodenal biopsies in CeD patients on GFD versus GD and normal healthy population also showed an alteration of gut microbiota. CeD patients on GD showed an increase in *Bacteroides-prevotella, Clostridium leptum, Histolitycum, Eubacterium, Atopobium* and decrease in *Bifidobacterium* spp., *B.longum*, *Lactobacillus* spp., *Leuconostoc*, *E. Coli* and *Staphylococcus* compared to the normal population [50,51,52,53,54]. When CeD patients were treated with GFD, the increased microbial concentration was reduced to that in the normal population, thus suggesting that diet influenced gut microbiota. However, most studies showed only partial restoration of the microbiota when CeD patients were put on a GFD [47,48,49]. In addition, some of these patients were symptomatic for CeD even on GFD and showed relative abundance of *Proteobacteria* and decreased number of *Firmicutes and Bacteroides* suggesting dysbiosis as a cause of persistent GI symptoms even on GFD [55]. The precise reason for the inability of GFD to restore the microbiota similar to healthy subjects is not well understood, but it can be speculated that this may be due to individual genetics or prebiotic effect of GFD [55,56,57]. Although no cause or effect relationship can be deduced from these studies, the consensus is that dysbiosis may contribute to CeD. They further showed that patients with Dermatitis Herpeteformis (DH) also had a characteristic gut microbiota, with increased *Firmicutes, Bacteriodes (Sterptococcus and Prevotella*) suggesting that gut microbiota may play a role in disease manifestation [58].

To understand the biochemical mechanism of the effect of gut microbiota in CeD, germ-free mice were colonized with bacteria from CeD and healthy subjects, respectively. In the germ-free mice, *Lactobacillus*, had a protective effect, while *Pseudomonas aeruginosa* was associated with CeD development [59]. *P. aeruginosa* was found to secrete LasB eleastase that altered intestinal barrier and facilitated translocation of gliadin peptides to the lamina propria where they activated the mucosal immune system. In contrast, *Lactobacillus* strains produced proteases that cleaved gluten into smaller peptides, which were less likely to be translocated to lamina propria, thus reduced their immunogenicity [59].

## 4. Factors Modulating Gut Colonization in Celiac Disease:

### 4.1. Association with HLA-Haplotypes, Breast Feeding, Birth, Antibiotic Exposure

There is a strong association between HLA-DQ2/8 haplotypes and CeD. Several investigators have examined this association with the gut microbiota. Infants with HLA-DQ2 and HLA-DQ8 and first-degree relatives with CeD have increased *Firmicutes* and *Proteobacteria* and less *Actinobacteria* and *Bifidobacterium*, suggesting that HLA genotype is associated with gut colonization by specific bacteria more prevalent in CeD patients and their relatives [20,60]. However, the HLA-DQ2/8 haplotype is also common in the general population, suggesting that genetics alone cannot explain the high prevalence of CeD.

In HLA-DQ2/8 haplotype infants, the gut microbiota was further affected by feeding type, with breastfeeding having a protective effect against CeD [61,62,63]. Breastfed babies had higher *Clostridium leptum, Bifidobacterium longum*, and *Bifidobacterium breve* compared to formula fed babies, whose colon had higher counts of *Bacteroides fragilis, Clostridium coccoides-Eubacterium rectale* and *E.coli.* Breastfeeding has is thought to have a protective effect on the development of CeD but could not be confirmed in some studies [64]. The bacteria acquired during birth and first few months of life have a significant effect on commensal organisms in gut. The adult gut microbiome is typically established by two years of age [65].

Observational studies have also shown an increased prevalence of CeD in children born by elective cesarean section (CS), with a negative association with vaginal delivery, and also with premature rupture of membranes, most likely related to possible gut dysbiosis. Babies born vaginally predominantly acquire bacteria from maternal vaginal and perianal flora. The gut microbiota of vaginally delivered infants is similar to their mother vaginal microbiota compared to elective CS infants who have reduced microbial diversity and fewer *Bifidobacterium* species [66,67].

Antibiotic use in the first year of life was also associated with intestinal dysbiosis, reduced fecal microbial diversity, and early onset of CeD [68,69]. Antibiotic-associated dysbiosis showed decreased numbers of *Bifidobacterium longum* and increased numbers of *Bacteroides fragilis* [70]. Moreover, Canova et al. demonstrated a dose-response relationship of antibiotic use with onset and risk of CeD, specifically with increased Cephalosporin intake [71]. As already discussed, CeD was associated with decreased *Bifidobacteria* counts, lending support to the hypothesis that dysbiosis is risk factor for celiac disease [43,44].

Environmental triggers, especially food processing and additives are becoming increasingly recognized as contributing factors to the rising incidence of CeD. Nanoparticles used in food processing, including metallic nanoparticles have antimicrobial activities. In vitro mouse model studies suggest an alteration in microbiota on exposure to these substances [72,73]. In mice, a dose dependent effect on the gut microbiome was noted with silver nanoparticles [74].

### 4.2. Effect of Gut Microbiota/Dysbiosis on Processing of Gluten

The effect on duodenal microbiota of the amount and timing of gluten introduction into the diet of an infant is controversial [75]. In small bowel partial digestion of gluten into peptides larger than ten amino acids are immunogenic, specifically 33-mer. Commensal microbiota, especially, *Lactobacilli* release peptidases that breakdown peptides and modify their immunogenic potential. *P. aeruginosa* is a pathogenic bacterium in patients with CeD. Caminero et al. demonstrated that *P. aeruginosa* is capable of enhancing immunogenicity of 33-mer peptide, while *Lactobacillus* species isolated from the non-CeD controls decreased the immunogenicity of the peptides produced by *P. aeruginosa* [59]. Gluten can be metabolized by 144 strains of 35 bacterial species [25]. Most of these strains were from phyla *Firmicutes* and *Actinobacteria,* bacteria that protect CeD. Herran et al. isolated 31 strains of gluten-degrading bacteria from the human small intestine, of which 27 strains demonstrated peptidolytic activity towards the 33-mer peptide [76]. *Lactobacilli* were the most representative genera, suggesting a protective role for *Lactobacillus* in gluten digestion with decreasing the immunogenicity of 33-mer peptide. To grow effectively, *Lactobacilli* require high amounts of amino acids for their nitrogen source and energy metabolism. *Lactobacilli* and *Bifidocbacterium* spp. are believed to have a complex proteolytic and peptidolytic system, which may be involved in breakdown of gluten and its peptides and have the potential to be used as a probiotic supplement in CeD patients [77].

### 4.3. Effect of Microbiota/Dysbiosis on the Intestinal Barrier in Celiac Disease

Intestinal defense against pathogens includes physical barrier created by a mucous and tight junction complexes between neighboring epithelial cells. These tight junctions impede entry of pathogens and toxic molecules across the gut wall. Sustained inflammation or infection can lead to deregulation in the expression of adhesion molecules at tight junctions leading to entry of microbes and toxic/immunogenic substances in lamina propria. Of several proteins involved in tight junctions, disassembly of zonulin has been implicated in CeD patients [26,27]. In vitro studies showed that zonulin can be induced by both gluten peptides and enteric bacteria [78]. Zonulin release in vivo has also been reported to be affected by changes in gut microbiota composition [79]. Some studies suggested that the gliadin peptides bound to pro-inflammatory cytokine CXCR3 receptor on the intestinal epithelium released zonulin, thereby disrupting tight junctions and increasing epithelial permeability [26]. CXCL10, a ligand for CXCR 3 was also overexpressed in the small intestine of CeD patients. CXCL10/CXCR3 axis can be activated by pathogens and has been suggested to play a role in initiating gluten-induced inflammatory processes in the small bowel [26]. Germ-free rats with triggering factors such as *Escherichia coli* CBL2 or *Shigella* CBD8 had significantly reduced numbers of goblet cells in the small bowel and altered intestinal barrier and tight junctions. However, when given gliadin and IFN-γ incubated with *Bifidobacterium bifidum* IATA-ES2, there was an increase in number of goblet cells, increased production of inhibitors of metalloproteinases and chemotactic agents, which provided a protective effect for the intestinal barrier [80]. Though these changes were established, it remains unclear if dysbiosis from CeD associated bacteria produced an inflammatory response to gluten or stimulated the mucosal inflammation response. Dysbiosis, through activation of the innate immune pathway may disrupt tight junctions/intestinal barrier and facilitate entry of incompletely digested gliadin peptides into the lamina propria. As discussed above, dysbiosis may also increase the amount and size of gliadin peptides due to differential peptidolytic activity of the gut microbiota.

### 4.4. Effect of Gut Dysbiosis on Mucosal Immunity in Celiac Disease

Microbiota colonization is necessary for the development and homeostasis of an optimal immune system. The gut microbial composition plays a role in regulation of the immune system. Alterations (dysbiosis or pathogenic organisms) may shift the immune response by favoring the development of certain subpopulation of lymphocytes that trigger a different cytokine response in the host. Physiologically the mucosal immune response to foreign antigens in the small intestine led to the development of tolerance to these antigens by apoptosis and active suppression by regulatory (Treg) T cells of antigen specific T cells [18,81,82,83,84]. As already discussed, there are similarities in activation of innate and adaptive immune system by immunogenic gliadin peptides and altered microbes in the gut. In patients with CeD, loss of tolerance to gluten is associated with activation of gluten specific CD4^+^ T cells in the lamina propria and upregulation of IL-15, a pro inflammatory cytokine [18,33,35,78,79,80]. The gut microbiota also plays a role in maturation of dendritic cells, macrophages in the small bowel and causes variation in interactions of gliadin peptides with CD4+T cells. Pathogenic bacteria activate the innate immune system through activation of TLR. TLR-4 and CD14 complexes recognize bacterial endotoxins and lipopolysaccharide and active the innate immune system to release proinflammatory cytokines. Soluble CD14 is a serum marker for activation of the innate immune system that increased in patients with untreated CeD, suggesting a role for dysbiosis in the pathogenesis of CeD. Altered gut microbiota can also activate Th1, Th2 and Th17 mediated immune responses similar to upregulation by gliadin peptides [33,85]. CeD associated bacteria can increase IL-17A which may be directed against it [82]. Further work is needed to better characterize gut microbiota changes in CeD, and their role in cytokine expression and clinical disease course.

These studies suggest that the gut microbiota affects gluten digestion, intestinal permeability, and the host immune system, all the mechanisms involved in pathogenesis of CeD. Although GFD can reduce the symptoms of CeD in most of the patients, however it does not completely restore the gut microbiota to that of healthy individuals [47,48,49]. Furthermore, dysbiosis was also observed in patients symptomatic on GFD. These studies suggest that gut dysbiosis contributes to the pathogenesis of CeD and utilization of probiotics may benefit CeD patients.

## 5. Probiotics in Celiac Disease:

Probiotics are live organisms, when ingested in adequate quantities provide a health benefit to the host [86]. They produce inhibitory substances that target pathogens, block their adhesion sites, compete for nutrients, prebiotics, degrade toxin receptors, and regulate immunity [87]. Dysbiosis, directly or indirectly contributes to CeD, therefore probiotics modulate the microbial profile of the duodenum and increase the beneficial colonizing microbes influencing the CeD prognosis. Several in vitro and clinical trials have been conducted to assess the use of probiotics.

In vitro studies have demonstrated that select *Lactobacilli* strains when added to sourdough fermentation, lyse the proline/glutamine-rich gluten peptides, reduce the gluten concentration to <10 ppm (gluten-free), and decrease their immunotoxicity. *Lactobacilli* strains from pooled probiotic culture, during simulated gastrointestinal digestion, hydrolyzed proline-rich synthetic peptides involved in CeD. Duodenal biopsies obtained from CeD patients following consumption of wheat bread produced with *Lactobacilli* showed no increase in IL2, IL-10, or IFN- levels compared to baseline [88]. Four strains of *Lactobacilli* (*L. ruminis, L. Johndoni, L. amylovorus, L. salivaris*) with highest gliadin peptide degrading activities reduced the immunotoxicity of gliadin peptides to induce CeD, were identified from the upper gastrointestinal tract of pigs [89]. In a study that challenged 20 CeD subjects to hydrolysed wheat gluten bread (containing *Lactobacillus alimentaris, L. brevis, L. sanfranciscenis, L. Hilgardi*) for six days, found no significant increase in IFN-γ compared to healthy controls [90]. Encouraging results were also obtained with in vivo studies when CeD patients in remission were challenged for 60 days with *Lactobacilli* predigested gluten. There was no worsening of symptoms, intestinal permeability or serological markers suggesting that *Lactobacilli* derived endopeptidase was capable of completely degrading gluten and reducing its toxicity for CeD patients [91]. These studies support the addition of probiotics rich in *Lactobacilli* spp. either to mitigate the effect of accidental or contaminant gluten exposure or for added benefits imparted by GFD.

Dysbiosis in CeD is associated with abnormal tight junction and increased intestinal permeability that lead to increased translocation of gliadin peptides to lamina propria. The De Simone Formulation, a probiotic mixture of mostly *Lactobacilli* and *Bifidobacteria*, not only completely hydrolysed -gliadin-derived epitopes 62–75 and 33-mer peptide, but also improved epithelial barrier function by stabilizing tight junctions [92]. Lindfors et al. studied the effects of probiotics including strains *Lactobacillus fermentum and B lactis* on human colon Caco2 cells and showed that *B. lactis* decreased intestinal permeability in a dose dependent manner [93]. Furthermore, a mixture of gliadin peptides and *Bifidobacteria* downregulated proinflammatory cytokines production from Caco cells [94]. Mononuclear cells treated with *B longum, B bifidum ES2* and then incubated with fecal samples from CeD patients found decreased proinflammatory cytokine production, suggesting that *Bifodobacterial* strains can reverse the effects of CeD associated bacteria [95].

Mouse models of CeD when challenged with gluten develops histological changes similar to CeD including intraepithelial lymphocytosis, villous atrophy, and crypt hyperplasia. These changes are associated with overexpression of CD71 (mediator of increased intestinal permeability), CD15, and IgA. When gluten digested with *Saccharomyces boulardi KK1* strain, was fed to mice, there was a decrease in CD 71 expression, and local cytokine production and reversal of histological changes, supporting the beneficial effects of probiotics [96]. *Lactobacillus rhamnosus GG* strain also decreased gliadin peptide induced changes in intercellular junction proteins and gliadin induced enteropathy in Wistar rats sensitized with IFN-. Similar beneficial effects of probiotic *B.longum* CECT 7347 were noted in small bowel of weaning animals sensitized with IFN-γ and fed gliadin suggesting that early administration of probiotics may have a protective effect on bowel mucosa [97]. The effects of different *Lactobacilli sp* on DQ8 transgenic mice showed modulation of innate and adaptive immune response [98,99]. They found that effects were strain specific. *B longum* NCC2705 strain produced a serine protease inhibitor (serpin) with immune modulating properties and prevented gliadin induced inflammation in genetically susceptible mouse model of CeD [98]. All these in vitro and animal studies demonstrated a beneficial effect of probiotics on digestion of gliadin peptides, intestinal barrier and immune system, which also showed beneficial effects on intestinal mucosa of mice [96,97,98,99,100,101,102,103]; Table 1.

Studies of the gut microbiota (fecal and duodenal biopsies) have revealed that *Lactobacilli* and *Bifidobacteria* reduce symptoms in CeD patients on a GFD, for a potential role either to mitigate the effect of accidental or contaminant gluten exposure [44,45,104]. These studies, along with in vitro and animal studies support the addition of probiotics to a GFD for beneficial or preventative effects. There are a few studies on the effect of probiotics in patients with CeD with cohorts ranging from 22–109 patients [105,106,107,108,109,110,111,112,113,114]. Most of these studies used *Bifidobacterium* strains, while a few studies used mixture of *Lactobacilli* spp. and *Bifidobacterium* spp. and one study used a probiotic preparation, the De Simone Formulation, made up of lactic acid and *bifidobacteria* (Table 2 and Table 3). Most of these studies demonstrated modulation of gut microbiota, decreased inflammatory cytokines causing reduction in CeD symptoms. In a double-blind, randomized, placebo-controlled study of 33 newly diagnosed pediatric patients with CeD on a GFD, were given *Bifidobacterium longum CECT* 7347 (*n* = 18) for three months. A reduction was observed in the intestinal inflammation and decreased sIgA in fecal samples, compared to the placebo group. It also improved growth metrics, affected peripheral lymphocyte subsets, and decreased TNF [106]. In a double-blind randomized placebo-controlled study in untreated CeD patients on a GFD (*n* = 12), *B infantis* (Natren Life Start, NLS) treatment for three weeks improved gastrointestinal symptoms but had no effects on diarrhea, intestinal permeability or serum levels (tTG serology), cytokines or chemokines [105]. This may be due to the limited duration of study, selection of probiotic, dose or symptomatic patients with Marsh 3b-3c histology on biopsy. However, *B infantis* decreased Paneth cells in duodenal mucosa and -defensin (antimicrobial peptide) suggesting its effect on modulation of the innate immune system [110].

*Bifidobacterium breve* BR03 and B63 given for three months reduced TNF production in children on a GFD suggesting an inhibitory effect on the innate immune system. However, the effect was lost three months after completing the study [107]. Using the same probiotic with a GFD in CeD adult patients, Primec et al. also showed a decrease in TNF-α, increase in *Fermicutes* spp. and decrease in *Verrucomicrobia* as well as new phyla [111]. TNF-α, an important mediator in activation of immune system has a direct and indirect effect on mucosal damage in CeD. Decreased cytokine production by probiotics along with GFD may have beneficial effect. These results need to be confirmed with larger prospective studies using appropriate dose and mix of probiotics. To study the effects of multispecies probiotics on irritable bowel syndrome (IBS-) type symptoms, Francavilla et al. enrolled 109 patients symptomatic CeD patients on GFD. Six weeks of treatment with probiotic showed decrease in severity of IBS-like symptoms compared to placebo treatment. This was associated with increase in anaerobes, *Bifidobacterium, Actinobacteria* in fecal samples [112].

In addition to endopeptidase abilities, probiotics can also alter the host immune function. *Saccharomyces boulardii* blocks toxin receptors, while *Bifidobacterium* and *Lactobacillus* strains secrete short chain volatile fatty acids, hydrogen peroxide, and antibacterial peptides (Lactocidin, Acidophilin, and Lactacin B). By reducing intraluminal pH, pathogenic bacteria were reduced. Furthermore, promoting epithelial growth factor (EGF) enhanced barrier function [115]. Certain strains, such as *Lactbacilli, GG, B. lactis*, and *Saccharomyces boulardii* regulate humoral modulation through expression of TGF, IL-10, and IL-6, which subsequently promote B-cell maturation in favor of IgA secretion.

There is a paucity of studies in probiotics in humans, and thus difficult to determine their role in the management of CeD [1,34,116,117,118,119]. In addition, the studies cannot conclusively demonstrate that probiotics improved gut barrier function, likely due to short duration or inadequate dose or strain, or their inability to modulate the gut microbiota. Further work is needed to better understand the role of probiotics in CeD.

## 6. Summary

Probiotics can influence the CeD by three potential mechanisms. The first is to digest the gluten proteins to non-immunogenic small polypeptides, eliminating and/or reducing the trigger for CeD, thus preventing its onset. The second is to maintain the intestinal barrier by preventing the access of immunogenic polypeptides to lamina propria. The third and most interesting is the role of probiotics in the homeostasis of the gut microbiome and regulation of both the innate and adaptive immune systems. Though alterations in the gut microbiota/dysbiosis are associated with the development of CeD, its exact function in pathogenesis remains unclear. The limited number of human studies show the positive effects of probiotics as a therapeutic modality in CeD, but more studies are needed specifically to modulate the gut microbiome to alter the disease course. To-date probiotics are unable to provide a durable modification of gut microbiota, and duodenal dysbiosis persisted. There are also concerns regarding safety of probiotics, including documented bacteria, lack of regulation, and lack of knowledge regarding interactions with the host microbiota [119]. There appears to be a role for probiotics to modulate the gut microbiota in CeD, however, further randomized studies, especially in humans, are needed to better understand its role in treating CeD. A basic understanding of the biochemical/molecular mechanism by which probiotics influence CeD will help to precisely formulate type and concentration of beneficial microbes to develop a safe therapeutic modality to alter the CeD course.

## Figures and Tables

**Table 1 nutrients-11-02375-t001:** Animal model studies to study the efficacy of probiotics in celiac disease.

Probiotic		Study Design	Major Findings	Advantages of Probiotics	Reference
Composition—Strain(s)	Duration of Administration				
*Lactobacillus casei* ATCC 9595	21 days	Transgenic mice expressing DQ-8 mucosally immunized, challenged with intra-gastric gliadin	Enhanced gliadin specific response mediated by CD4^+^T cells, gliadin specific IFNγ expression, pro-inflammatory polarisation of cytokine milieu, no enteropathy like mucosal response	Inherent adjuvancy of *L.casei* can be used to enhance both mucosal and T-cell mediated responses.	D’Arienzo et al. 2008. [101]
*Lactobacillus casei* ATCC 9595	35 days	Transgenic mice expressing DQ, -mucosally immunized, challenged with intra-gastric chymotrypsin digested gliadin	Complete recovery of villous blunting, decreased weight loss, recovered basal TNF-α levels and no change in CD25^+^ cells and levels of IL-2.	*L. casei* was effective in rescuing the normal mucosal architecture and Gut associated lymphoid tissue homoeostasis in a mouse model of gliadin –induced enteropathy.	D. A’Arienzo et al. 2011 [102]
*Saccharomyces boulardi* KK1 strain, hydrolysed the 28-kDa gliadin fraction	30 days	BALB/c mice –three generations fed with gluten free diet to develop gluten sensitivity (G-), immunised with 50 µg whole gliadin emulsified in Freund’s adjuvant. The probiotic *S. boullardi* KK strain or control were administered orally for seven days and then fed gluten diet for 30 days. Oral administration of microbes continued twice per week. Intestine samples collected one day after the last dose.	The G+ mice developed villous atrophy crypt hyperplasia, and infiltration of T cells, inflammation and over expression of CD71. *S. boulardi* administration improved enteropathy development, decreased epithelial cell expression of CD71 and localized cytokine production.	Anew mouse model for human CD based on histopathological features and common biomarkers. *S. boulardi* a probiotic to treat CeD by reversing disease development.	Papista et al. 2012 [96]
*Bifidobacterium longum* CECT 7347	Ten days from birth.	Newborn rats fed gliadin alone or sensitized with IFN-α and then fed gliadin.	In sensitized animals *B. longum* administration increased NFκB expression and IL-10 but reduced TN- α expression, and CD4+ and CD4+/Fox3+ cell populations and increased CD8+ T cell populations, contrary to the results without probiotic treatment.	*B. longum* regulates inflammatory cytokine production and CD4+ T cell mediated immune response in an animal model of gliadin induced enteropathy.	Laparra et al. 2012 [103]
*Lactobacillus rhamnosus* GG (L.GG) ATCC 53103	Ten days after birth, L.GG was administered for 10 days.	Newborn Wistar rats divided into four groups, Ctrl (without treatment); PTG (sensitized with 1000 U IFN-γ intraperitoneally after birth and administered gliadin for 10 days); L.GG treated with L.GG for 10 days); Co-administered (sensitized and L.GG together); Pre-treated (sensitized, then pre-treated with gliadin and then administered with L.GG for 10 days).After treatments the animals were sacrificed and jejunal tissue samples were collected.	Probiotic strain L.GG increased expression of genes related to tight junctions TJ) and adherin junctions (AJ), after gliadin induced damage and symptoms of CeD.	Probiotic L.GG protects rat intestinal mucosa damage and can be developed for the therapeutic management of gluten-related disorders in humans.	Orlando et al. 2018 [100]

**Table 2 nutrients-11-02375-t002:** Probiotics influence Celiac Disease development and treatment in adults.

Probiotics		Trial		Outcome	Conclusions	Reference
Composition	Duration	Country	Participants/Design			
*Bifido bacterium* natren life start (NLS)	Three weeks—s2 week run-in, three weeks treatment, and follow up on day 50	Argentina	Placebo controlled, double blind study; 22 participants; 12 participants received the probiotic capsule, 10 placebo.	Effect of NLS on: (i) intestinal permeability; (ii) outcome of clinical symptoms by GSRS questionnaire; (iii) modification of immunologic indicators influenced by gluten.	Administration of NLS to untreated CeD patients does not modify protein abnormalities but might improve symptoms and produced some immunologic changes. Participant pool was small, more trials are needed.	Smecuol et al. 2013 [105]
A proprietary blend of 450 billion viable lyophilized bacteria (9 strains) known as the De Simone formulation, previously VSL#3.	12 weeks study	Australia	47 enrolments, final results were for 42 participants; equally divided in the active group that received the probiotics and 21 in the placebo group.	Primary outcome: microbial counts of and comparison between baseline and end of study of predominant, pathogenic and opportunistic bacteria. Secondary efficacy outcomes: urinary metabolomics and faecal lactoferrin	No significant change in the gastrointestinal microbial counts in CeD individuals with persistent symptoms over 12 weeks period. Future studies to increase the dosage of VSL#3 and duration of treatment.	Harnett et al. 2016 [109]
Yogurt with probiotic from PIA, Nova Petropolis-RS (undetermined microbial concentration)	30 days	Brazil	17 healthy and 14 participants with celiac disease	Faecal bifidobacteria concentration after consuming 100 g of yogurt in the morning.	Faecal bifidobacteria was higher in healthy patients compared to CeD patients. Probiotic yogurt consumption increased the bifidobacteria number in CeD patients, but could not reach the concentration in healthy participants.	Martinello et al. 2017 [113]
*Bifidobacterium infantis* Natren Life Start super strain (NLS-SS)	Six weeks	Argentina	41 participants, in three groups: (i) *n* = 24, CeD active, no treatment; (ii) *n*=12, CeD active with NLS; (iii) *n* = 5; CeD 1 year GFD	Determine mucosal expression of innate immune markers: number of macrophages, Pancth cells and α-defensin-5 expression by immunohistochemistry in duodenal biopsies.	Duodenal biopsies revealed that *B. infantis* decreased all the three markers, macrophage counts, Pancth cell counts and α-defensin-5 in CeD patients. However, the decrease in macrophage counts was higher in gluten free diets. Future studies are needed to study methods to obtain synergistic effect of GFD and *B.infantis* supplementation.	Pinto-Sanchez et al. 2017 [110]
A product containing five strains: *Lactobacillus casei, Lactobacillus plantarum, Bifidobacterium animalis* subsp. Lacti, *B. breve* Bbr8 LMG P-17501 and *B. breve* B110 LMG P-17500.	A six-week treatment period, precede by 2-week run in period followed by a 6 week follow up phase for a total of 14 weeks.	Italy	Prospective, double- blind, randomized placebo-controlled parallel group study. 109 participants were included in the study. 54 in the probiotic and 55 in the placebo group.	Primary outcome to determine if probiotics improve GI symptoms as assessed by Irritable Bowel syndrome severity scoring system (IBS-SSS). Five secondary outcomes including modification in gut microbiota and metabolic fecal profile.	Probiotics significantly decreased the IBS-SSS and GSRS scores compared to the placebo. Presumptivr lactic acidbacteria, *Staphylococus* and *Bifidobacterium* counts were also higher with probiotic treatment compared to the placebo group. Six-week Probiotic treatment was effective in managing IBS-type symptoms. Probiotics in CeD patients on strict GFD diet modified the gut microbiota positively by increasing the *Bifidobacteria*.	Francavilla et al. 2019 [112]

**Table 3 nutrients-11-02375-t003:** Probiotics influence Celiac Disease development and treatment in children.

Probiotics		Trial		Outcome	Conclusions	Reference
Composition	Duration	Country	Participants/Design			
*Bifidobacterium longum* CECT 7347	3 months	Spain	33 participants in a double blind, randomized, placebo-controlled trial.	Baseline and post-intervention outcomes included immune phenotype of peripheral blood cells, serum cytokine concentration, fecal secretory IgA content, anthropometric parameters and intestinal microbiota composition.	Probiotic treatment showed greater height percentile, decreased peripheral CD3^+^ T lymphocytes, and slightly reduced TNF-α concentration. The number of *Bacteroides fragilis* and content of secretory IgA in the stool was also reduced by the probiotic treatment. The small sample size is a limitation of the study.	Olivares et al. 2014 [106]
*Bifidobacterium breve* BRO3 and *B. breve* B632	Three months	Slovenia	Double blind placebo-controlled trial with 49 participants randomized into two groups: First group of 24 received the probiotic and the second group of 25 received the placebo. 18 healthy children were included as controls.	Outcomes: Serum production of interleukin 10 (IL-10); tumor necrosis factor alpha (TNF-α).	TNF-α levels decreased after 3 months of probiotic treatment, however on follow up after 3 months, the levels increased. The IL-10 levels were below detection.	Klemenak et al. 2015 [107]
*Bifidbacterium breve strains B632 and BRO3*	3 months	Slovenia	Double-blinded, placebo-controlled study with 40 CeD patients and 16 healthy children. The CeD patients were in teo groups of 20 each, with one receiving the probiotic and the other placebo.	Determination of microbiome after probiotic treatment.	Probiotic treatment increased the *Firmicutes* and restored the physiological *Firmicute/Bacteriodetes* ratio. Three-month administration of probiotic can restore the microbiota of CeD patients similar to healthy children.	Quagliariello et al. 2016 [108]
*Bifidbacterium breve strains B632 and BRO3*	3 months	Slovenia	Double-blinded, placebo-controlled study with 40 CeD patients and 16 healthy children. The CeD patients were in teo groups of 20 each, with one receiving the probiotic and the other placebo.	To study the influence of probiotics on the fecal microbiome, Short chain fatty acids (SCFA) and serum TNF-α.	*Verrucomicrobia, Paracubacteria* and some yet unknown phyla of bacteria and archaea showed a strong correlation to CeD. These new microbiota may have a role in CeD.	Primec et al. 2019 [111]
*Lactobacillus reuteri; Lactobacillus rhamnosus* and some unidentified	USA, Finland, Germany and Sweden	Different time periods	A prospective study using a cohort of 6520 genetically susceptible children. 1460 children were reported probiotic use in the first year of life.	To study the association between the exposure of probiotics via dietary supplements or by infant formula by the age 1 year to the development of celiac disease autoimmunity (CDA) or CeD.	Overall exposure of probiotics during the first year of age was not associated with CDA or CeD. However, intake of probiotics via dietary supplements was associated with increased risk of CDA.	Uusitalo et al. 2019 [114]

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
