# Peer review of "The Gut Microbiota in Celiac Disease and probiotics"

_nutrients, 2019, doi:10.3390/nu11102375_

Round 1

Reviewer 1 Report

The manuscript is a extensive review of the gut microbiota, celiac disease and potential use of probiotics.
This is a well written manuscript and I have only a very few comments.

Most of the comments refers to the references.

Page 4, rows 154 - 156: I would like you to add some references regarding the statement of cesarean sections and risk of CD. 

Page 5, row 240: Please add the reference for the WHO definition of probiotics that you use in the sentence.

Page 6, rows 248 - 251: There is something wrong with this sentence. Francavilla et al has not written the paper that is referenced in the text. Check the references.

Page 6, row 266: I recommend to delete the reference here, it is mentioned again at the end of the same sentence.

Page 7, row 347: I dont think that this is the correct reference?  

Author Response

Page 4, row 154-156: Ref 66,67 Page 5 - WHO definition.  Food and Agriculture Organization and World Health Organization Expert Consultation.  Evaluation of health and nutritional properties and powder milk and live lactic acid and bacteria.  Cordoba, Argentina: Food and Agriculture Organization of the United Nations and World Health Organization; 2001.   Should be reference 87.   Page 7 - Pace LA and Crowe SE.  Complex relationships between food, diet, and the microbiome.  Gastroeneterol Clin N Am 45 (2016): 253-265.  

Reviewer 2 Report

In this review authors Chibbar et al. have discussed the exclusively about Celiac disease, and the role of gut microbiota and probiotics. 

The paper is well-written. A few comments:

Understanding gluten is the major cause for the disease, what is the role of other food items which can trigger the disease such as emulsifiers, additives etc..,?  As gut microbiome composition is also affected with the food, what is the role of age, sex, concomitant medications in the CED? A quick illustration on mechanisms (pathology & probiotics roles in modulation of the microbiome) may add value to the paper. Adding a table on different clinical and mouse models will definitely add value to the paper.

Author Response

A section has been added regarding food additives; however, this is very topical and a subject for another in depth review, beyond the scope of the current manuscript.  

The suggestion of adding tables to summarize the results of clinical studies based on animal and human trials has been accepted, and has been incorporated.  

Reviewer 3 Report

Minor issues with English usage - will be detected on proof reading. Nil else. 

Author Response

Thanks for your comments.